# Effect of Intermittent Fasting on Cardiometabolic Health in the Chinese Population: A Meta-Analysis of Randomized Controlled Trials

**DOI:** 10.3390/nu16030357

**Published:** 2024-01-25

**Authors:** Katy Cheung, Vicky Chan, Stephanie Chan, Martin Ming Him Wong, Gary Ka-Ki Chung, Wai-Yin Cheng, Kenneth Lo, Fangfang Zeng

**Affiliations:** 1JC School of Public Health and Primary Care, The Chinese University of Hong Kong, Hong Kong 999077, Chinagchung@cuhk.edu.hk (G.K.-K.C.); 2Department of Food Science and Nutrition, The Hong Kong Polytechnic University, Hong Kong 100872, China; vickycwk.chan@polyu.edu.hk (V.C.); wai-yin-nano.cheng@polyu.edu.hk (W.-Y.C.); 3Institute of Epidemiology and Health Care, University College London, London WC1E 6BT, UK; ming.wong.23@ucl.ac.uk; 4Research Institute for Future Food, The Hong Kong Polytechnic University, Hong Kong 100872, China; 5Research Institute for Smart Ageing, The Hong Kong Polytechnic University, Hong Kong 100872, China; 6Department of Public Health and Preventive Medicine, School of Medicine, Jinan University, No. 601 Huangpu Road West, Guangzhou 510632, China

**Keywords:** cardiometabolic indicators, intermittent fasting, time-restricted eating, weight loss

## Abstract

The efficacy of intermittent fasting (IF), as an emerging weight management strategy, in improving cardiometabolic health has been evaluated in various populations, but that among Chinese individuals has not been systematically studied. A comprehensive search on multiple databases was performed to identify eligible randomized controlled trials (RCTs) up to October 2022. The primary outcome was post-intervention weight loss, and secondary outcomes included changes in cardiometabolic indicators. Effect estimates were meta-analyzed using a random-effects model. In total, nine RCTs with 899 Chinese participants were included. Time-restricted eating was the most adopted IF protocol in this study (six out of nine), followed by alternate-day fasting. The IF intervention significantly reduced body weight, body mass index, body fat mass, homeostatic model assessment of insulin resistance, low-density lipoprotein cholesterol, and triglycerides when compared with control groups. However, no statistically significant reductions in waist circumference, total cholesterol, high-density lipoprotein cholesterol, fasting glucose, systolic blood pressure, and diastolic blood pressure were found. To sum up, IF can be a weight management strategy and may improve the cardiometabolic health of Chinese adults, but more long-term trials using different IF strategies are required to generate robust evidence of its efficacy.

## 1. Introduction

In the past decades, the rate of general and abdominal obesity has increased markedly among Chinese adults [1]. In 2019, more than 50% adults aged 18 or above were classified as overweight or obese according to body mass index (BMI) [2]. The number of Chinese people with overweight or obesity is estimated to grow to 790 million in 2030 [3]. As an established risk factor of cardiometabolic diseases, the obesity epidemic has increased the burden on the medical system in China [4]. By 2030, the medical cost attributed to overweight and obesity in China is predicted to be USD 61 billion [3]. Furthermore, at the same BMI values, Chinese individuals are at higher risk of cardiometabolic diseases than Black and White adults in the US [5]. Developing effective strategies for obesity prevention and management is important in the Chinese population.

Various forms of fasting have been developed, including intermittent fasting (IF), calorie restriction, long-term fasting, very low-calorie diet, and fasting-mimicking diet [6]. In recent years, IF has been an emerging dietary intervention for weight management. IF refers to diverse recurring eating patterns that require fasting for different time periods, ranging from hours to a few days [7], which includes alternate-day fasting (ADF), periodic fasting, and time-restricted eating (TRE). Previous studies demonstrated the potential benefits of IF with respect to obesity, insulin resistance, dyslipidaemia, hypertension, and inflammation [8,9].

Although IF is a potential weight management strategy, its efficacy may vary by dietary habits in the study population because of the adherence issue. A recent study reported that the average eating window of 1596 Chinese adults was 13.03 h, in which 45% of individuals commenced eating at 8:00–9:00 a.m., and 52% of participants ceased eating at 8:00 p.m. [10]. According to NHANES 2009–2014, the average eating window, first mealtime, and last mealtime of 15,341 adults aged 20 or above were 12.2 ± 0.06 h, 8:08 a.m., and 8:18 p.m., respectively [11]. With a potentially longer eating duration in the Chinese population than in the US, conducting IF by restricting the eating window may not have the same efficacy. In addition, whether IF is beneficial mainly because of calorie restriction is unclear. A one-year randomized controlled trial (RCT) with 139 Chinese participants found that early TRE (8:00 a.m. to 4:00 p.m.) combined with daily calorie restriction had similar effects as calorie restriction only on weight loss and body fat reduction in people with obesity [12], but more trials in the Chinese region are warranted to verify the finding.

In brief, the previous literature suggested a lack of systematic review of the efficacy of IF in improving cardiometabolic health of the Chinese population and whether the potential benefit of IF is independent of calorie restriction during the intervention. Gaining additional understanding on the efficacy of IF in the Chinese population may help health care professionals formulate evidence-based weight loss plans for individuals who are overweight and obese.

## 2. Materials and Methods

This meta-analysis was performed according to the Preferred Reporting Items for Systematic Reviews and Meta-Analyses 2020 statement [13]. The study protocol was registered in PROSPERO (CRD42022359891).

### 2.1. Search Strategy

Five databases, namely, Ovid MEDLINE, Embase (via Ovid), Ovid Emcare, WanFang, and the China Academic Journal Network Publishing Database (CNKI) were used to identify relevant RCTs that investigated the efficacy of IF on Chinese populations up to October 2022, using a combination of keywords related to IF (e.g., intermittent fasting or time-restricted feeding or periodic fasting) and RCT (e.g., RCT or clinical trial or controlled clinical trial or randomization or random assignment). Reference lists of the included studies were also retrieved for potential eligible studies. The details of the search terms that were used are summarized in Appendix A.

### 2.2. Study Selection

Two independent reviewers (KC and KL) were involved in the two-stage process for the selection of studies. After duplicate studies were excluded, the titles and abstracts were first screened. The abstracts followed by full contents of studies that met the pre-specified eligibility criteria were assessed with reference to the inclusion and exclusion criteria.

### 2.3. Inclusion Criteria

RCTs evaluating the effects of IF on improving the cardiometabolic health of the ethnic Chinese population aged 18 years old or above, without restriction on the sex and health condition of the participants.Intervention with duration more than 3 weeks to capture the treatment effects of long-term duration only, with reference to a previous meta-analysis of RCTs that examined the efficacy of nutritional intervention on weight loss and body composition [14].

### 2.4. Exclusion Criteria

Observational studies, review papers, comments, letters, news, notes, protocols, papers, or abstracts from conference proceedings.Studies without an abstract or full text in English or Simplified/Traditional Chinese.Studies without a control group.

### 2.5. Data Extraction

Data extraction was performed independently by two reviewers (KC and VC). The following study characteristics were extracted from each included study: (a) first author, (b) publication year, (c) location where the trial was conducted, (d) age and sex of participants, (e) sample size, (f) participants’ weight and BMI at baseline, and (g) details of the trials (e.g., intervention duration, control group, IF methods adopted, outcomes, adverse events). Outcomes that appeared in at least three included studies were extracted to perform meta-analysis. For multiple post-intervention assessments, data from the assessments immediately after the intervention was extracted. If different units were used to report the same parameter in the studies, then the most frequently adopted unit was used for consistency. The reviewers reached a consensus on studies to be included before starting data analysis. Cases of inconsistencies were resolved via thorough discussions.

### 2.6. Quality and Risk-of-Bias Assessment

The guidelines from the Cochrane Handbook for Systematic Reviews of Interventions were followed for risk-of-bias assessment. The six domains of bias being assessed are selection bias, performance bias, detection bias, attrition bias, reporting bias, and other bias. The two reviewers (KC and KL) conducted the assessment independently and rated each item as low, unclear, or high risk. Funding of the included studies was assessed in terms of other bias. Studies that received industrial or commercial funding were rated as high risk [15].

### 2.7. Outcomes

IF interventions including ADF, 5:2 fasting, and daily TRE were evaluated. The IF interventions could be applied at different time intervals. The primary outcome was absolute weight loss after intervention. With reference to a previous meta-analysis of RCTs that examined cardiometabolic outcomes [16], cardiometabolic parameters, including BMI, waist circumference (WC), body fat mass (BFM), total cholesterol (TC), high-density lipoprotein-cholesterol (HDL-C), low-density lipoprotein cholesterol (LDL-C), triglycerides (TG), fasting plasma glucose (FPG)/fasting glucose (GLU), homeostatic model assessment of insulin resistance (HOMA-IR), systolic blood pressure (SBP), and diastolic blood pressure (DBP), were selected as secondary outcomes.

### 2.8. Statistical Analysis

Cochrane Collaboration Review Manager (version 5.4) was used for statistical analyses. Mean differences (MDs) were used to express the changes of outcomes between the pre- and post-intervention, with corresponding 95% confidence intervals (CIs). A *p* value of <0.05 was considered statistically significant. For studies with multiple IF interventions, the more commonly adopted intervention groups among other included studies were involved in the meta-analysis. For outcomes that were reported by at least three included studies, we stratified the analysis according to the control groups (ad libitum diet or calorie-restricted diet) to examine the relative advantage of the IF intervention over the calorie-restricted diet and also when IF is compared with the ad libitum diet. Random-effects model was adopted to combine outcomes from included studies and incorporate heterogeneity among studies [15]. The statistical heterogeneity of the included studies was estimated using Cochran’s Q test and *I*^2^ statistics. Heterogeneity was described as moderate (*I*^2^ > 30%), substantial (*I*^2^ > 50%), and considerable (*I*^2^ > 75%). For studies identified as having high heterogeneity, input data were double-checked [15].

Sensitivity analysis was completed by excluding each included study one by one and recalculating the pooled estimate of the remaining studies to identify the studies that largely affected the summary effect or heterogeneity. For studies with multiple IF interventions, we recalculated the effect estimates by using the less commonly adopted intervention arms.

## 3. Results

### 3.1. Study Selection

The flow of study search, screening, and selection is described in Figure 1. A total of 746 articles were identified from the initial database search. After title and abstract screening, full text screening was conducted on 224 articles, and nine RCTs with 899 participants whose nationality were Chinese (excluding subjects in intervention groups irrelevant to the review) were included in the meta-analysis [12,17,18,19,20,21,22,23,24].

### 3.2. Study Characteristics

The study characteristics of the nine RCTs included in the meta-analysis are summarized in Table 1. In general, three trials evaluated 16:8 TRE [12,21,23], one evaluated 14:10 TRE [19], two studies evaluated the 5:2 diet [20,22], one evaluated ADF [24], and two trials evaluated both 16:8 TRE and ADF [17,18]. Among them, three RCTs conducted interventions with multiple arms [17,18,23]. The follow-up period of trials ranged from 3 weeks to 12 months. Forest plots were not stratified by ad libitum and calorie-restricted groups, including both types of control, due to the limited number of studies (with fewer than three studies reporting the same outcome).

### 3.3. Effect on Anthropometric Measurement

Pooled analysis of all included studies with a total of 749 participants showed that adopting IF resulted in statistically significant reductions in body weight when comparing the IF intervention group with both control groups (ad libitum diet: MD = −2.61 kg, 95% CI = −3.37 to −1.85, *I*^2^ = 76%) (calorie-restricted diet: MD = −1.40 kg, 95% CI = −1.60 to −1.20, *I*^2^ = 0%). The overall MD was also statistically significant (MD = −2.20 kg, 95% CI = −2.75 to −1.66, *I*^2^ = 87%) (Figure 2).

BMI analysis that involved 693 participants from eight studies also showed statistically significant reductions in comparison to both control groups (ad libitum diet: MD = −1.37 kg/m^2^, 95% CI = −2.00 to −0.73, *I*^2^ = 91%) (calorie-restricted diet: MD = −0.55 kg/m^2^, 95% CI = −0.92 to −0.18, *I*^2^ = 0%). The overall MD is also statistically significant (MD = −1.07 kg/m^2^, 95% CI = −1.67 to −0.48, *I*^2^ = 93%) (Appendix A).

Similarly, IF resulted in a statistically significant reduction in WC (MD = −4.04 cm, 95% CI = −7.09 to −0.99, *I*^2^ = 65%) relative to the ad libitum diet control group. However, the result with the calorie-restricted diet control group was not statistically significant (MD = −0.42 cm, 95% CI = −1.46 to 0.62, *I*^2^ = 0%). The overall result with 536 participants from six studies was not statistically significant (MD = −2.12 cm, 95% CI = −4.30 to 0.07, *I*^2^ = 83%) (Appendix A).

### 3.4. Effect on Body Fat Composition

Amongst 408 participants from four studies, some showed that IF could lead to a statistically significant reduction in BFM (MD = −1.55 kg, 95% CI = −1.66 to −1.43, *I*^2^ = 0%) (Appendix A). The consistency of the results was not affected despite the differences in body fat assessment methods.

### 3.5. Effect on Blood Lipid Profile

Eight studies involving 712 participants reported MDs in lipid profiles, including TC, HDL-C, LDL-C, and TG. No statistically significant differences in TC were found when compared with both control groups and the overall MD (ad libitum diet: MD = −1.43 mg/dL, 95% CI = −4.19 to 1.33, *I*^2^ = 33%) (calorie-restricted diet: MD = 2.69 mg/dL, 95% CI = −2.85 to 8.23, *I*^2^ = 0%) (overall: MD = −0.63 mg/dL, 95% CI = −3.20 to 1.94, *I*^2^ = 30%) (Appendix A). A similar trend was found for HDL-C (ad libitum diet: MD = −0.92 mg/dL, 95% CI = −2.00 to 0.15, *I*^2^ = 37%) (calorie-restricted diet: MD = 0.88 mg/dL, 95% CI = −0.70 to 2.45, *I*^2^ = 0%) (overall: MD = −0.43 mg/dL, 95% CI = −1.29 to 0.44, *I*^2^ = 31%) (Appendix A). Although the overall reduction was statistically significant for LDL-C, the result was insignificant when compared with that of the calorie-restricted control group (ad libitum diet: MD = −4.11 mg/dL, 95% CI = −7.66 to −0.57, *I*^2^ = 66%) (calorie-restricted diet: MD = −0.01 mg/dL, 95% CI = −3.59 to 3.58, *I*^2^ = 0%) (overall: MD = −2.86 mg/dL, 95% CI = −5.52 to −0.21, *I*^2^ = 56%) (Appendix A). A statistically significant overall reduction in TG could be observed when compared with both control groups (ad libitum diet: MD = −2.22 mg/dL, 95% CI = −4.19 to −0.24, *I*^2^ = 33%) (calorie-restricted diet: MD = −5.94 mg/dL, 95% CI = −6.28 to −5.59, *I*^2^ = 0%) (overall: MD = −3.35 mg/dL, 95% CI = −6.13 to −0.58, *I*^2^ = 97%) (Appendix A).

### 3.6. Effect on Blood Glucose

Data on FPG/GLU were reported in eight studies with 712 participants. A statistically significant reduction in FPG/GLU was observed when IF was compared with the ad libitum diet control group (MD = −8.42 mg/dL, 95% CI = −13.96 to −2.88, *I*^2^ = 89%), but not for the calorie-restricted diet control group (MD = 1.65 mg/dL, 95% CI = −0.95 to 4.25, *I*^2^ = 0%). The overall reduction was not statistically significant (MD = −4.61 mg/dL, 95% CI = −10.08 to 0.86, *I*^2^ = 93%) (Appendix A). The study of Zheng et al. was not included in the pooled analysis because no exact figure of the MD, SD of MD, CI, and *p*-value was reported. A statistically significant reduction in HOMA-IR was observed (MD = −0.48, 95% CI = −0.80 to −0.17, *I*^2^ = 51% (Appendix A), which included 417 participants from five studies.

### 3.7. Effect on Blood Pressure

Data on SBP and DBP were reported in five studies with 351 participants. IF did not lead to a statistically significant reduction in SBP (MD = −1.99 mmHg, 95% CI = −4.19 to 0.21, *I*^2^ = 0%) (Appendix A) and DBP (MD = −1.84 mmHg, 95% CI = −4.64 to 0.96, *I*^2^ = 56%) (Appendix A).

### 3.8. Risk-of-Bias Assessment

The risk-of-bias assessments for the included studies are summarized in Figure 3. Six studies described how a random sequence was generated, and three studies described how allocation concealment was achieved. For performance bias, seven and two studies were rated to have high and unclear level of bias, respectively. Seven studies were considered to have a high risk of blinding participants and personnel (performance bias) because masking the participants to their intervention was not possible. Two studies stated that none of the staff responsible for outcome measurements were informed of the assignment of participants and were rated with an unclear level of performance bias. The outcomes were detected entirely using machines, which is why the detection bias was low for all of the studies. For attrition bias, despite cases that were lost to follow-up, the attrition rate was relatively low for all of the studies. The reporting bias for all of the studies was uncertain given the lack of clarity on whether any measured, but not reported, outcomes based on the results only and without a trial protocol would be produced. None of the included studies indicated that their funding was from industrial or commercial sponsors, but three studies did not mention their sources of funding. Funnel plots to evaluate publication bias were not used in view of the small number of included studies (less than 10), where an appropriate level of the power of the tests to distinguish chance from real asymmetry could not be guaranteed [25].

### 3.9. Safety

Three studies reported the occurrence of mild adverse events (AE) such as dizziness, nausea, and constipation during the IF intervention [12,22,24]. One trial did not mention any AE [21]. No severe AEs were reported in the remaining five included studies [17,18,19,20,23]. Details of reported AEs are listed in Appendix A.

### 3.10. Sensitivity Analysis

Sensitivity analysis was performed for each outcome. For most of the study outcomes, the statistical significance and heterogeneity remained similar regardless of the study being excluded. The following paragraphs summarize the substantial changes in heterogeneity and statistical significance after performing the leave-one-out approach. We also summarize the sensitivity analysis results of body weight and BMI, the primary outcomes, in Appendix A. For body weight, after Cai et al. was excluded [17], *I*^2^ decreased from 87% to 68%. Apart from this result, the results changed from significant to insignificant among the calorie-restricted diet subgroup, which could be due to the large sample size. More studies with the calorie-restricted diet as the control group should be performed to verify the significance of the findings. For BMI, after Che et al. were excluded [19], the *I*^2^ reduced from 93% to 70% possibly because of the relatively larger sample size and weighting of each individual study.

Three included studies were three-armed [17,18,23], which is why data of another intervention group were selected for meta-analysis in the sensitivity analysis (Appendix A). The second intervention group of Cai et al. and Chair et al. was ADF, which was less adopted among the other included studies. For Xie et al., the two intervention groups were early TRE (eTRE) and mid-day TRE (mTRE), and eTRE was included in the meta-analysis because it was suggested to be more effective than later TRE in improving insulin resistance, glycemic metabolism, and blood pressure, with reference to a previous network meta-analysis [26]. When the mTRE group was included instead of the eTRE group, or when the ADF group was included instead of the TRE group, neither the statistical significance nor the direction of intervention effects changed substantially (Appendix A).

Given that TRE was the most adopted IF protocol in this work, further analysis was performed to compare TRE with the ad libitum diet or the calorie-restricted diet by removing non-TRE studies. For TC, when Guo et al. and Liu et al. were excluded [20,22], *I*^2^ decreased from 30% to 8%, and the results changed from insignificant to significant (MD: −2.35 mg/dL [95% CI: −3.90, −0.81]). For LDL, when Guo et al. and Liu et al. were excluded [20,22], the results changed from significant to insignificant (MD: −1.36 mg/dL [95% CI: −4.04, 1.33]), while no substantial change in heterogeneity was observed. For HOMA-IR, when Guo et al. was removed [20], no substantial change in heterogeneity was observed, but the results changed from significant to insignificant (MD: −0.43 [95% CI: −0.87, 0.01]). For DBP, when Guo et al. and Liu et al. were excluded [20,22], the results changed from insignificant to significant (MD: −3.58 [95% CI: −7.09, −0.07]), while no substantial change in heterogeneity occurred.

## 4. Discussion

### 4.1. Summary of the Main Findings

The pooled analysis of nine RCTs that compared IF with ad libitum diet or calorie-restricted diet indicated that participants that engaged in IF showed significant reductions in body weight and BMI when compared with participants receiving the ad libitum diet and calorie-restricted diet. The IF intervention also reduced BFM, HOMA-IR, LDL-C, and TG significantly when compared with control groups. Meanwhile, no statistically significant overall reductions appeared in WC, TC, HDL-C, FPG/GLU, DBP, and SBP in the analyses. While the effect of IF on TG and LDL outcomes was statistically significant, the magnitude may be modest.

Similar to results of previous studies, IF did not lead to a significant change in blood glucose level but improved insulin sensitivity [27,28,29,30]. Insulin sensitivity indicates how the human body responds to insulin and regulates blood glucose level. Impaired insulin sensitivity will lead to insulin resistance, which is associated with metabolic syndromes (MetS) such as hypertension and dyslipidemia [31]. Energy depletion is achieved through IF, and this will lead to prolonged reduction of insulin secretion, which activates AMP-activated protein kinase and improves insulin sensitivity and glucose homeostasis by promoting glucose uptake and utilization [32]. Blood glucose level did not change significantly as a result of IF, possibly because of the mechanisms involved in maintaining blood glucose level and preventing hypoglycemia, namely, glycogenolysis and gluconeogenesis [33].

Moreover, IF improves the lipid profile by boosting fatty acid oxidation through nuclear expression of peroxisome proliferator-activated receptor alpha (PPARa) and peroxisome proliferator-activated receptor-gamma coactivator 1-alpha (PGC1a) and increasing apolipoprotein modulation in the liver [34]. Hepatic triglyceride accumulation was reduced because of increased fatty acid oxidation, thereby reducing very low-density lipoprotein (VLDL) production, followed by reduced serum levels of VLDL, low-density lipoprotein (LDL), and small and dense LDL (sdLDL) [34]. The apolipoprotein B level also decreased as it was one of the structural components of VLDL, LDL and sdLDL, while the apolipoprotein A level increased, which led to increasing the HDL level in the blood as it was the precursor of HDL [34]. This condition may explain why IF could reduce LDL-C and TG instead of the HDL-C level.

In the present meta-analysis, IF resulted in a significant reduction in body weight, which was consistent with that of previous studies [35,36,37]. BMI was calculated based on weight and height only, which is why the BMI value would decrease considerably when body weight decreased [38]. Moreover, waist circumference is an indicator of abdominal adiposity [37]. Previous evidence suggested that IF may have favorable effects on waist circumference reduction [9,18,39]. However, no significant results were observed in this meta-analysis, possibly because of the short duration of the IF intervention. The study duration of this meta-analysis ranged from 3 weeks to 12 months, and only one out of nine included studies that lasted for 12 months, which hindered its efficacy in long-term health outcomes. In the previous review, under prolonged fasting, the serum level of leptin decreased, whereas that of adiponectin did not change, which promoted fatty acid oxidation with an anti-inflammatory effect, and adiponectin was suggested to have an inverse correlation with visceral adiposity [40]. However, in exchange for short-term fasting, no change in the serum leptin level occurred [40]. Therefore, a short duration of the IF intervention may impact abdominal adiposity.

We also examined whether the statistically significant findings from meta-analysis may inform clinical practices. For body weight, at least 5% weight loss from baseline was considered clinically significant and could reduce the risk of developing type 2 diabetes and cardiovascular disease [41]. For the current analysis, the overall reduction in body weight after the IF intervention was statistically significant (MD = −2.20 kg, 95% CI = −2.75 to −1.66, *I*^2^ = 87%), and the average weight loss after the IF intervention was 4.8%, which was close to the clinically significant value. Every 2 kg/m^2^ incremental increase in BMI among the Chinese population was associated with an elevated relative risk of 15.4% for coronary heart disease, 6.1% for total stroke, and 18.8% for ischemic stroke [42]. Although the overall reduction in BMI after the IF intervention was statistically significant (MD = −1.07 kg/m^2^, 95% CI = −1.67 to −0.48, *I*^2^ = 93%), the finding might not be clinically significant. In addition, the pooled level of LDL-C reduction in the present meta-analysis was only −2.86 mg/dL, which was lower than the clinically significant value, i.e., each 38.7 mg/dL (1 mmol/L) lowering in LDL-C reduced the risk of cardiovascular mortality as demonstrated by another meta-analysis [43]. However, our observed results have not been consistent enough to propose recommendations for clinical practice, given limited studies and the heterogeneity in participants’ characteristics such as their comorbidities. In this meta-analysis, most of the included studies were conducted among patients with diverse medical conditions (i.e., one study on non-alcoholic fatty liver disease [NAFLD], three on overweight or obese, two on prediabetes or diabetes, one on MetS, and one on spinal cord injury), and only one study was performed among healthy people. With reference to a previous meta-analysis, a statistically significant reduction in TC was observed among healthy people (WMD = −6.41 mg/dL, 95% CI = −9.64 to −3.18, *I*^2^ = 76.8%), but not in people with a disease history including NAFLD, type 2 diabetes mellitus, and MetS (WMD = −13.23 mg/dL, 95% CI = −37.70 to 11.23, *I*^2^ = 79.3%) [44]. Also, a higher TC reduction was observed among female participants (WMD = −18.71 mg/dL, 95% CI = −29.64 to −7.78, *I*^2^ = 89.7%) than among male participants [44]. This finding implies that IF intervention effects could be affected by medical conditions and sex. Future studies controlling potential confounders are of vital importance to verify the health effects of IF among the Chinese population.

### 4.2. Potential Mechanisms of Different IF Strategies

While IF may exert health benefits via calorie restriction [39,45], additional potential mechanisms of how IF strategies benefit human health are still under discussion. For example, ADF was able to increase free fatty acid oxidation and deplete amino acids periodically, which accelerated adipose tissue lipolysis and hepatic amino acid uptake for gluconeogenesis. Thus, ADF was suggested to be beneficial in cardiovascular health and helpful for weight loss [46]. The 5:2 diet could reduce fasting serum insulin acutely during a two-day fast and moderately stimulate the release of adiponectin by adipose tissues, which improved insulin sensitivity and lowered the risks of developing cancer, heart disease, or diabetes [47]. TRE might improve oscillations in circadian clock gene expression and reset the molecular mechanisms of energy metabolism through a restricted eating window, which was beneficial for weight management [48,49]. Also, TRE was suggested to be able to alter the microbial community composition (i.e., increased the ratio of Firmicutes to Bacteroidetes) that resides in the intestinal tract and lower gut permeability, which lowered the risks of developing gut diseases [49]. One of the included studies in the present meta-analysis, in addition to the beneficial cardiometabolic effects, discussed the metabolic pathways of gut microbiota. Several gut microbial metabolites, namely, lipopolysaccharide (LPS), short-chain fatty acids (SCFAs), and trimethylamine *n*-oxide (TMAO), play an important role in immunity and inflammation, which closely correlate with cardiovascular health [20]. After the IF intervention, plasma SCFAs were elevated from 4.97 ng/mL (95% CI: 4.09 to 5.86) to 6.14 ng/mL (95% CI: 4.90 to 7.37), while plasma LPS reduced from 156.1 ng/L (95% CI: 69.1 to 243.1) to 74.6 ng/L (95% CI: 36.5 to 112.7; *p* = 0.011), and no change in plasma TMAO was observed [20]. Authors have proposed that IF altered the gut microbiota and improved gut microbiota homeostasis, which was highly associated with improving cardiometabolic risk factors [20].

This meta-analysis indicates that IF is useful in weight loss. However, it is not beneficial to some cardiometabolic parameters, such as SBP and DBP. A previous meta-analysis with 694 participants showed similar findings, in which TRE could significantly reduce SBP (MD = −4.15 mmHg, 95% CI = −6.73 to −2.30; *p* < 0.0001) but not DBP [50]. In another one-year observational study with 1422 participants, regardless of sex, significant reductions of SBP (from 131.6 ± 0.7 mmHg to 120.7 ± 0.4 mmHg; *p* < 0.001) and DBP (from 83.7 ± 0.4 mmHg to 77.9 ± 0.3 mmHg; *p* < 0.001) were observed and were greater among people who fasted for longer [51]. From a physiological perspective, IF may trigger blood pressure reduction by increasing parasympathetic activity with brain-derived neurotrophic factor production, promoting renal Na excretion, and improving sensitivity of natriuretic peptides and insulin receptors [51]. Only four RCTs were included for the analysis because of the limited studies. Therefore, further studies should be conducted to verify the effects of IF on blood pressure among the Chinese population.

### 4.3. Additional Benefits of IF Compared with the Calorie-Restricted Diet

Another controversial issue is the additional weight loss effects of IF when compared with the calorie-restricted diet. Unlike people who are on a calorie-restricted diet, people adopting IF are not necessarily required to limit their calorie intake [52]. However, IF emphasizes the importance of staying in sync with natural circadian rhythms instead of restricting total calorie intake [52]. A previous meta-analysis with 205 participants with obesity reported that IF was more beneficial than the calorie-restricted diet for weight loss (SMD = −0.21 kg, 95% CI = −0.40 to −0.02; *p* = 0.028) but no difference was observed in BMI reduction (SMD = 0.02 kg/m^2^, 95% CI = −0.16 to 0.20; *p* = 0.848) [39]. Another review suggested that IF and the calorie-restricted diet were equally effective in body weight loss (IF: 4–8% vs. calorie-restricted diet: 5–8%) and fat mass reduction (IF: 11–16% vs. calorie-restricted diet: 10–20%), but IF outperformed the calorie-restricted diet in lean mass retention [53]. Moreover, an earlier meta-analysis suggested that IF was comparable to the calorie-restricted diet (MD = −0.26 kg, 95% CI = −0.31 to 0.84; *p* = 0.37) and could be a reliable weight loss alternative [54]. The current meta-analysis pooled the effects from multiple studies conducted in China and showed statistically significant overall reductions in body weight, BMI, and TG when compared with the calorie-restricted diet and the ad libitum diet. While more large-scale studies with a longer study duration should be conducted, the current findings demonstrate the potential clinical significance of IF in cardiometabolic health, which warrants in-depth investigation.

### 4.4. Adherence to IF Strategies in China

Weight loss is important in reducing the risk of various cardiometabolic diseases [55]. Hence, good adherence to a weight loss strategy is crucial.

In China, the eating habits of the Chinese population are a potential barrier that lowers people’s acceptance and adherence to IF. The average eating window of Chinese adults is 13.03 ± 2.02 h per day [10]. People with fixed-time jobs or living in urban areas have busy schedules and commonly have irregular meals [10]. Moreover, midnight snacks are a common food culture in southern China and Hong Kong [56]. Chinese individuals who are accustomed to having many dinner gatherings and have a habit of enjoying midnight snacks may find it difficult to adopt IF. Hence, the significance of evening gatherings with family or friends should be taken into consideration to improve IF adherence in the Chinese population [57].

In this meta-analysis, six studies reported IF adherence rates ranging from 84% to 97.5%, which suggested that IF is feasible in the short term. However, good adherence to IF in the long term could not be concluded due to small sample sizes (39 to 174) and varied study durations (5 weeks to 12 months). Also, in one study, participants were not blinded, so they might have already been interested in TRE, which resulted in high adherence rates [23].

### 4.5. Adverse Events of IF in the Chinese Population

Although IF is gaining popularity, studies reporting the safety of IF strategies are insufficient. Common adverse events include vertigo, nausea, insomnia, headaches, and exhaustion [58]. In the current meta-analysis, no serious adverse events were disclosed. However, people with chronic diseases such as diabetes, heart failure, asthma, and cancers are suggested to be more likely to experience adverse events [58].

### 4.6. Limitations of the Study

This meta-analysis has some limitations that should be noted for cautious interpretation. First, most included RCTs had a small number of recruited subjects and a short follow-up period, with only one study lasting for 12 months [12]. Second, different IF methods were pooled for the meta-analysis because of the limited data for comparison with two control groups (ad libitum diet and calorie-restricted diet). The comparison between IF and the calorie-restricted diet has not been comprehensively assessed. Third, considerable heterogeneity was observed in multiple meta-analysis results, which may limit the interpretation accuracy. The between-study differences, such as adherence to fasting regimes, metabolic status, diet, and exercise level of the subjects, which were rarely reported in the included studies, might also contribute to the high heterogeneity [59]. The number of eligible studies in this work was very limited, and the results had considerable heterogeneity. Thus, the findings of recent studies may not be consistent enough to inform clinical recommendations. Age, sex, intervention duration, and morbidity status were considered potential sources of heterogeneity in previous meta-analysis, and meta-regression analyses were performed to explore the heterogeneity [54,60,61]. However, different studies obtained inconsistent results. For example, Pascual et al. concluded that intervention duration did not have a confounding effect [54], while Jahrami et al. revealed that fasting duration was a confounding factor, but age and sex were not [61]. Faris et al. suggested that diet, fasting duration, body weight and health status before fasting, and exercise level were considerable interfering factors [60]. More future studies on the effect of IF among the Chinese population with different characteristics and morbidities should be conducted to explore the heterogeneity and investigate how these factors are associated with the effect of IF interventions [15].

## 5. Conclusions

Among Chinese adults, adopting IF for at least three weeks can result in some degree of weight loss and improvement in cardiometabolic health. IF may have potential roles in weight management, but more well-designed studies are necessary to provide robust evidence of its effects on weight management by considering the effects of sex and meal timing and to account for the dietary habits of the Chinese population. Moreover, the possible adverse incidents and long-term effects of IF interventions need to be studied by conducting more long-term and large-scale clinical trials.

## Figures and Tables

**Figure 1 nutrients-16-00357-f001:**
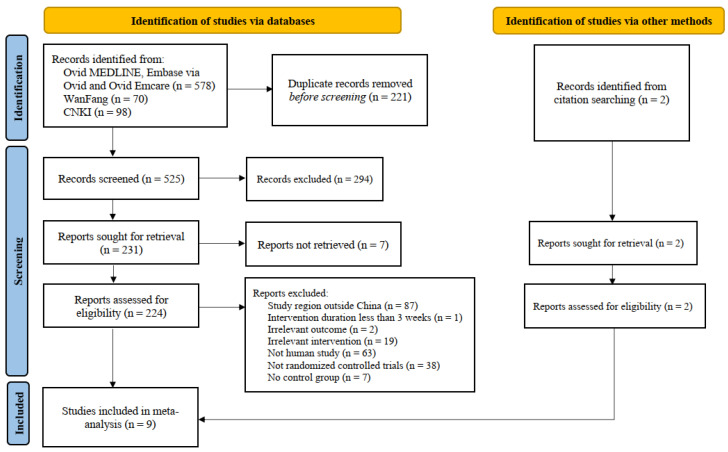
Flow diagram of study select.

**Figure 2 nutrients-16-00357-f002:**
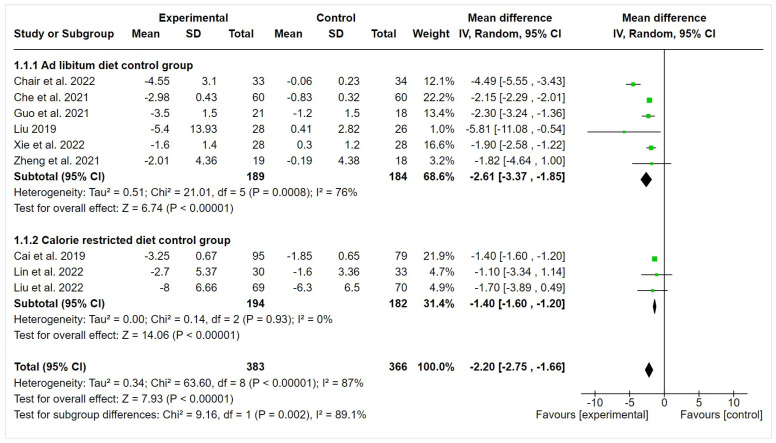
Forest plot for differences in body weight (in kg) between intermittent fasting and control groups [12,17,18,19,20,21,22,23,24]. Size of the green boxes represented study weight while the black diamond represented the pooled effect.

**Figure 3 nutrients-16-00357-f003:**
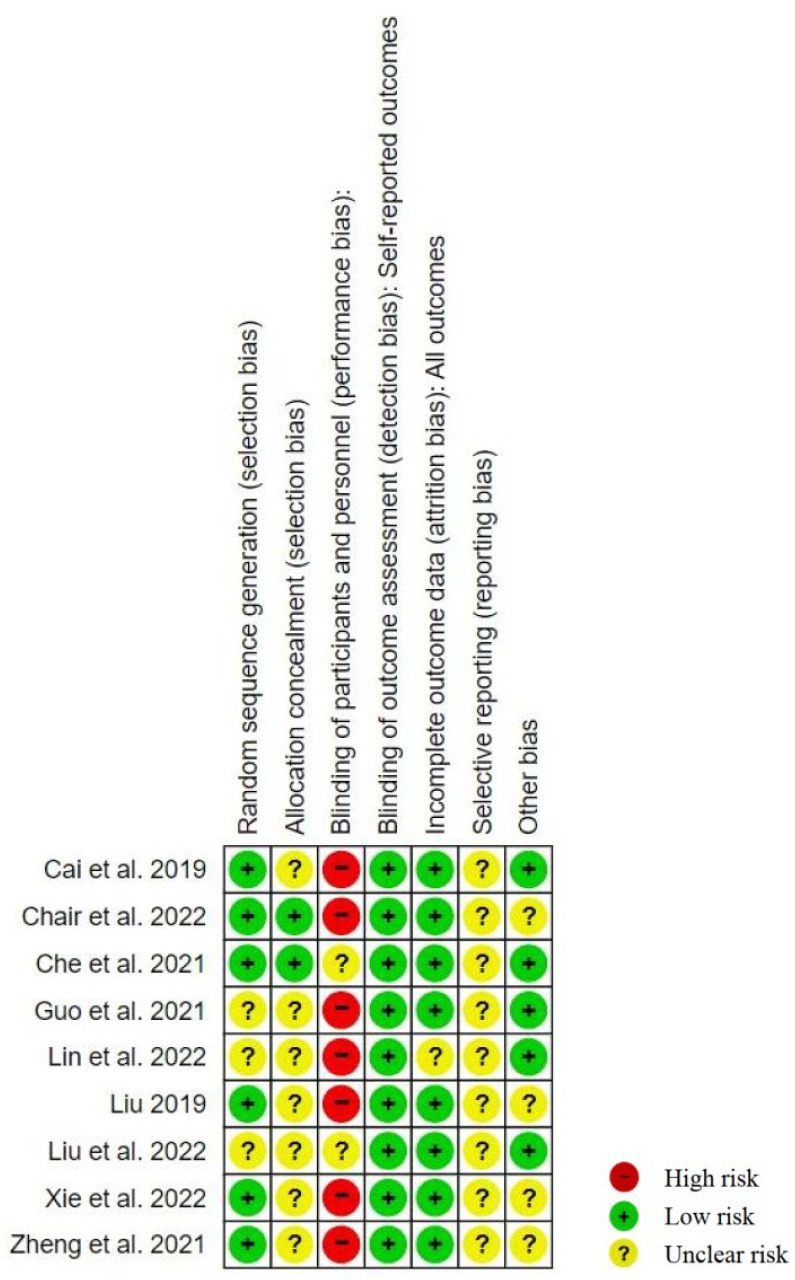
Risk-of-bias summary of the included studies [12,17,18,19,20,21,22,23,24].

**Table 1 nutrients-16-00357-t001:** Study characteristics of nine included studies.

Author, Year	Study Location	Follow-Up Period	Subject (*n*)	Comorbidities of Participants	IF Method	Comparator	Age ± SD	Gender (% of Male)	Baseline Weight ± SD (kg)	Weight Change (%)	Baseline BMI ± SD (kg/m^2^)	Other Outcomes	Reported Adverse Events	Compliance to IF
Cai et al., 2019 [17]	Changsha, Hunan Province	12 weeks	264	Patients with non-alcoholic fatty liver dis	(i) 16:8 TRE (freely arranged) (*n* = 95) (ii) ADF (*n* = 90)	Consumed 80% of energy needs every day (*n* = 79)	Intervention: (i) TRE: 33.56 ± 6.23 (ii) ADF: 35.50 ± 4.42 Control group: 34.54 ± 6.96	33.0	Intervention: (i) TRE: 74.98 ± 8.02 (ii) ADF: 75.32 ± 8.53 Control: 72.94 ± 8.00	Intervention: (i) TRE: −4.4% (ii) ADF: −5.4% Control: −1.6%	Intervention: (i) TRE: 26.76 ± 1.59 (ii) ADF: 26.12 ± 2.21 Control: 26.34 ± 2.73	WC, BFM, TC, HDL-C, LDL-C, TG, GLU	No severe adverse events	Intervention: 97.5%
Chair et al., 2022 [18]	Changsha, Hunan Province	3 weeks	101	Overweight and obese individuals with prediabetes	(i) 16:8 TRE (freely arranged) (*n* = 33) (ii) ADF (*n* = 34)	Eating ad libitum (*n* = 34)	Intervention: (i) TRE: 36.06 ± 7.67 (ii) ADF: 34.68 ± 4.37 Control: 34.97 ± 6.23	36.7	Intervention: (i) TRE: 74.98 ± 8.02 (ii) ADF: 75.78 ± 8.46 Control: 72.15 ± 8.47	Intervention: (i) TRE: −6.1% (ii) ADF: −6.4% Control: −0.1%	Intervention: (i) TRE: 26.74 ± 1.58 (ii) ADF: 26.46 ± 2.36 Control: 26.47 ± 1.84	WC, TC, HDL-C, LDL-C, TG, GLU	No severe adverse events	Not available
Che et al., 2021 [19]	Tianjin	12 weeks	120	Overweight patients with type 2 diabetes	14:10 TRE (08:00–18:00) (*n* = 60)	Eating ad libitum (*n* = 60)	Intervention: 48.21 ± 9.32 Control: 48.78 ± 9.56	54.2	Intervention: 75.06 ± 4.42 Control: 74.68 ± 4.35	Intervention: −4.0% Control: −1.1%	Intervention: 26.42 ± 1.96 Control: 26.08 ± 2.14	TC, HDL-C, LDL-C, TG, FPG, HOMA-IR	No severe adverse events	Intervention: >6 days per week
Guo et al., 2021 [20]	Dongguan, Guangdong Province	8 weeks	39	Patients with metabolic syndrome	5:2 diet (*n* = 21)	Eating ad libitum (*n* = 18)	Intervention: 40.2 ± 5.7 Control: 42.7 ± 4.1	53.8	Intervention: 77.8 ± 13.6 Control: 74.1 ± 8.6	Intervention: −4.5% Control: −1.6%	Intervention: 28.0 ± 7.8 Control: 27.7 ± 2.3	WC, BFM, TC, HDL-C, LDL-C, TG, GLU, HOMA-IR, SBP, DBP	No severe adverse events	Intervention: 91.3% Control: 78.3%
Lin et al., 2022 [21]	Taiwan	8 weeks	63	Middle-aged perimenopausal women with BMI > 24 kg/m^2^ or WC > 80 cm	16:8 TRE (10:00–18:00 or 12:00–20:00) (*n* = 30)	Daily low-calorie diet of 1400 kcal (*n* = 33)	Intervention: 50.1 ± 7.5 Control: 54.2 ± 7.9	0	Intervention: 65.9 ± 9.7 Control: 65.8 ± 8.8	Intervention: −4.1% Control: −2.4%	Intervention: 25.9 ± 3.7 Control: 25.7 ± 3.8	WC, TC, HDL-C, LDL-C, TG, GLU, HOMA-IR, SBP, DBP	Not mentioned	Intervention: 84%
Liu, 2019 [22]	Nanchang, Jiangxi Province	20 weeks	54	Overweight and obese people	5:2 diet (*n* = 28)	Eating ad libitum with moderate exercise (*n* = 26)	Intervention: 43.04 ± 8.75 Control: 41.69 ± 8.56	31.5	Intervention: 72.96 ± 10.94 Control: 72.59 ± 8.26	Intervention: −7.4% Control: +0.6%	Intervention: 27.41 ± 2.52 Control: 27.94 ± 1.85	WC, TC, HDL-C, LDL-C, TG, FPG, SBP, DBP	Decreased concentration, thirsty, dizziness, low blood sugar level, abdominal bloating, constipation, unstable emotion	Not available
Liu et al., 2022 [12]	Guangzhou, Guangdong Province	12 months	139	Patients with obesity	16:8 TRE (08:00–16:00) (*n* = 69)	Daily calorie restriction by 75% (*n* = 70)	31.9 ± 9.1	51.1	88.2 ± 11.6	Intervention: −9.0% Control: −7.2%	Intervention: 31.8 ± 2.9 Control: 31.3 ± 2.6	WC, BFM, TC, HDL-C, LDL-C, TG, GLU, SBP, DBP, HOMA–IR	No deaths or serious adverse events, occurrences of mild adverse events were similar in the two groups	Intervention: 84.0% Control: 83.8%
Xie et al., 2022 [23]	Beijing	5 weeks	82	Healthy individuals without obesity	Early 16:8 TRE (06:00–15:00) (*n* = 28) Mid-day 16:8 TRE (11:00–20:00) (*n* = 26)	Eating ad libitum (*n* = 28)	Intervention: (i) eTRE: 28.68 ± 9.71 (ii) mTRE: 31.08 ± 8.44 Control: 33.57 ± 11.60	22.0	Intervention: (i) eTRE: 61.1 ± 8.8 (ii) mTRE: 61.0 ± 11.7 Control: 61.2 ± 9.9	Intervention: (i) eTRE: −2.6% (ii) mTRE: −0.3% Control: +0.5%	Intervention: (i) eTRE: 22.7 ± 3.1 (ii) mTRE: 21.4 ± 2.2 Control: 21.5 ± 2.9	BFM, FPG, HOMA-IR	No severe adverse events	Intervention: 96.8%
Zheng et al., 2021 [24]	Chengdu, Sichuan Province	8 weeks	37	Patients with spinal cord injury	ADF (*n* = 19)	Eating ad libitum (*n* = 18)	35.76 ± 6.92	91.9	Intervention: 61.06 ± 6.29 Control: 61.44 ± 5.29	Intervention: −3.3% Control: −0.3%	Intervention: 21.88 ± 1.59 Control: 22.34 ± 1.88	--	Nausea, dysphoria, pulmonary infection, hypoglycemia, hunger, irritability	Not available

Abbreviations: ADF, alternate-day fasting; BFM, body fat mass; BMI, body mass index; DBP, diastolic blood pressure; eTRE, early time-restricted eating; FPG, fasting plasma glucose; GLU, fasting glucose; HDL-C, high-density lipoprotein cholesterol; HOMA-IR, homeostatic model assessment of insulin resistance; IF, intermittent fasting; LDL-C, low-density lipoprotein cholesterol; mTRE, mid-day time-restricted eating; SBP, systolic blood pressure; SD, standard deviation; TC, total cholesterol; TG, triglycerides; TRE, time-restricted eating; and WC, waist circumference.

## Data Availability

The data presented in this study are available on request from the corresponding authors.

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
