# Peer review of "Effect of Intermittent Fasting on Cardiometabolic Health in the Chinese Population: A Meta-Analysis of Randomized Controlled Trials"

_nutrients, 2024, doi:10.3390/nu16030357_

Round 1

Reviewer 1 Report

Comments and Suggestions for Authors Dear Editor of Nutrients – MDPI   Thank you for inviting me to be a Referee for this scientific article: “Effect of Intermittent Fasting on Cardiometabolic Health in  Chinese Population: A Meta-analysis of Randomized Controlled Trials”. - Despite my opinion transcription regarding what is requested in the electronic form Nutrients – MDPI, I would like to add the following comments:   Lines 18-32: Abstract - Brief but completely in accordance with what has been scientifically evaluated. Lines 35-72: Introduction: with well-defined content and adequate citations   In general, the article is well written and meets the requirements of a Meta-Analysis study. It also takes into important consideration the following aspects: - Lines 67-71: The initial question was well posed “previous literature suggested a lack of systematic review of the efficacy of IF on improving cardiometabolic health of Chinese population, and whether the potential benefit of IF is independent of calorie restriction during intervention. Through gaining additional understanding on the efficacy IF in Chinese population, it may help healthcare professionals to formulate evidence-based weight loss plans for individuals with overweight and obesity.”   - Materials and Methods: are explicit and correctly described (Search strategy, Study selection, Inclusion/exclusion criteria, Data Extraction,Quality and risk-of-bias, Outcomes, Statistical analysis): - lines 75-77 “This meta-analysis was performed according to the Preferred Reporting Items for Systematic Reviews and Meta-Analyses 2020 statement [12]. And it was registered correctly “The study protocol has been registered in the PROSPERO (CRD42022359891).   - Lines 79-81 “Five databases, namely Ovid MEDLINE, Embase (via Ovid), Ovid Emcare, WanFang and China Academic Journal Network Publishing Database (CNKI) were used to identify relevant RCTs investigating the efficacy of IF on Chinese populations up to October 2022,]”   - Lines 92-99: the “Inclusion criteria: Randomized controlled trials evaluating the effects of IF on improving cardiometabolic health of the ethnic Chinese population aged 18 years old or above, without restriction on sex and health condition of the participants” are well defined. Intervention with duration more than 3 weeks to capture the treatment effects of long-term duration only…” Exclusion criteria: …”Studies without a control group”.   - The authors used data from studies with IF in accordance with that published by Yang F et al (2021) [(Yang F Liu C, Liu X, Pan X, Li X, Tian L, Sun J, Yang S, Zhao R, An N, Yang X, Gao Y and Xing Y (2021)]Effect of Epidemic Intermittent Fasting on Cardiometabolic Risk Factors: A Systematic Review and Meta-Analysis of Randomized Controlled Trials. .2021.669325 ], that is (ADF and/or “Time-restricted feeding (TRF) Complete fast (no calories) for at least 12 h a day, and eating freely the rest of the time; the 16:8 fasting pattern currently prevails ( 9, 16, 18)”.   - Lines 112-113: “To perform meta-analysis, outcomes appeared in at least three included studies were extracted.”…   - Lines 146-155: Heterogeneity and sensitivity analysis clearly explained.   In fact, the 9 studies included meet all the requirements defined in the materials and methods chapter. However, there are some aspects that deserve changes that are referenced in red: (in red purposals to be potentially changed and also to be praised).   Lines 107-117: I believe it is important that the Data Extraction also includes the type of population included with regard to their characteristics/morbidities: examples: overweight and obese adults with prediabetes (Chair et al); overweight patients with type 2 diabetes (Che et al); in patients with metabolic syndrome (Guo et al); healthy individuals without obesity (Xie et al); patients with spinal cord injury (Zheng et al); non-alcoholic fatty liver disease patients (Cai et al)...This aspect must be considered and a new column must be included in Table S2 with these morbidities as they could be a reason to explain the considerable heterogeneity found in several clinical parameters.   Results - Lines 165-169: “3.2. Study characteristics. The study characteristics of the 9 RCTs included in the meta-analysis were summarized in Table S2. In general, five trials evaluated 16:8 TRE [11,16,17,20,22], one evaluated 14:10 TRE [18], two studies have evaluated 5:2 diet [19,21], and one has evaluated ADF [23]. Among them, 3 RCTs conducted interventions with multiple arms [16,17,22].” As already mentioned, Table S2 will be enriched if it has another column with the characteristics of the participants included (see above). Line 168: “and one has evaluated ADF [23]”. According to Chair et al (2022) [17]: “The participants were randomized into the ADF group (n = 34), 16/8 TRF group (n = 33), and control group (n = 34). Therefore, there are 2 studies that evaluated ADF and not just one, that of Zheng et al (2021) [23]   Lines 192-195: “3.4. Effect on body fat composition. The analysis on BFM which involved 408 participants from 4 studies showed that IF could lead to statistically significant reduction (MD = -1.55 kg, 95% CI = -1.66 to -1.43, p < 194 0.001, I2 = 0%; Figure S4)”; How was Body Fat Composition assessed/quantified in different studies? In the same manner in all studies?   Lines 196-219: “Effect on blood lipid profiles…Effect on blood glucose”; There are apparently positive results, such as those seen in the Forest plot in Figure S7 for differences in low density lipoprotein cholesterol (in mmol/L): …in relation to these results, it is suggested to convert the units mmol/L into mg/dL (for example, the reduction of 0.31 mmol/l in LDL (p<0.001), when comparing the Intermittent Fasting intervention with the Ad libitum diet, corresponds to a reduction of 12.0 mg/dL which clinically seems to be more intuitive of a possible benefit (because as demonstrated in the “meta-analysis study found that each 38.7 mg/dL (1 mmol/L) lowering in LDL-C reduced the risk of cardiovascular mortality (RR, 0.85; 95% CI, 0.81–0.89), (in: Khan SU, Michos ED. Cardiovascular mortality after intensive LDL-Cholesterol lowering: Does baseline LDL-Cholesterol really matter? American Journal of Preventive Cardiology. 2020).   - The same is suggested for the conversion of mmol/L to mg/dL of the Units referring to the values of total cholesterol, HDL, Triglycerides and glucose.   - Lines 173-230: The results transcribed/transferred to the text are consistent with those shown in the Forest plot of Figures S1 to S12 (and I expect that these Figures appear in the final publication). However, it is very important to discuss some aspects contained in them that can improve the discussion chapter, especially regarding the topic of the heterogeneity of the studies included (one of the most relevant aspects in this article): these will be the results of this meta-analysis relevant to establish/propose recommendations for clinical practice such as the considerable clinical and methodological heterogeneity found (even considering the sensitivity study and subgroup analysis used?   From reading the articles included in the Meta-Analysis (indicated below), some questions/suggestions emerge. Therefore, the studies highlight the following aspects (in red). - Chair et al (2022): these authors state - “A randomized controlled trial was conducted on a sample of 101 overweight and obese adults with prediabetes. The participants were randomized into the ADF group (n = 34), 16/8 TRF group (n = 33), and control group (n = 34). The intervention lasted for 3 weeks. Data on body weight, body mass index, waist circumference, blood glucose, and lipid profile were collected at baseline, at the end of the intervention, and at the 3-month follow-up….ADF group were instructed by a dietitian to consume 600 kcal on fasting days and to consume a usual diet on eating day… Purposes: n weight loss, blood glucose, and lipid profile in overweight and obese adults with prediabetes”. Why were the 33 individuals who performed Alternate-Day-Fasting not included in Table S2? (n=34) to better understand the numbers related to the ages in the intervention group (for example)…Cholesterol, glucose…. because it allows for a better discussion about it” the findings of this study revealed that ADF achieved more-significant reductions in body weight, BMI, and waist circumference than following a usual diet, suggesting that ADF may be a potentially effective weight-loss strategy – Chair et al “In this study, an ADF with a 24-hour interval was compared with 16/8 TRF , with results showing that ADF had a more significant effect on weight loss in terms of body weight and BMI reduction than 16/8 TRF”… Lines 269-272:” Since three included studies were three-armed [16,17,22], data of another intervention group was selected for meta-analysis in sensitivity analysis. The second intervention group of Cai et al. and Chair et al. was ADF, which was less adopted amongst other included studies”. There is no certainty about this statement by the authors… (not sufficient explanation to exclude the data)…

- Che et al: (2021). these authors state “time-restricted feeding on glycaemic regulation and weight changes in overweight patients with type 2 diabetes over 12 weeks”; “fed freely from 8:00 to 18:00 and fasted from 18:00 to 8:00 daily (a 14-h fast)” – no questions/suggestions to ask.   - Guo et al (2021): these authors state “cardiometabolic risk factors and the gut microbiota in patients with metabolic syndrome (MS); “adults with MS, ages 30 to 50 years”; “a 2-day fasting dietary schedule (the IF group, which involved a 75% of energy restriction for 2 nonconsecutive days a week and an ad libitum diet the other 5 days) for 8 weeks. “; “Cardiometabolic risk factors including body composition, oxidative stress, inflammatory cytokines, and endothelial function were assessed at baseline and at 8 weeks. The diversity, composition, and functional pathways of the gut microbiota, as well as circulating gut-derived metabolites, were measured” - obviously these are several different outcomes that can interfere with the discussion, especially on the topic of clinical heterogeneity, which is well demonstrated in the other outcomes studied…and can be used in a more enlarged discussion “In addition to the beneficial cardiometabolic effects … “More than 70 genes shifted after the intervention. Four of these pathways were predicted to be significantly upregulated in samples of the IF group (FDR…” our study demonstrates that short-term 2-day IF improves levels of adipokines, prevents lipid peroxidation, and improves vascular endothelial function among populations with MS, and the effects appear to be associated with alternations in gut microbiota composition, microbial-related metabolites, and activated metabolic pathways in the gut microbiome.”   - Regarding data from the study by Xie et al (2022): “randomized trial to compare the effects of the two TRF regimens in healthy individuals without obesity”; Why were only 28 participants considered when in fact N=28 + N=26 were included; why only eTRF? More favorable results? and which? It's because? (“five-week trial and were analyzed (28 in eTRF- food intake restricted to the early part of the day (eating during a period of no more than 8 h between 06:00 and 15:00, and fasting for the rest of the day), 26 in mTRF - food intake restricted to the middle of the day), 28 in control groups… eating during a period of no more than 8 h between 11:00 and 20:00, and fasting for the rest of the day)). The primary outcome was the change in insulin resistance…” we show that eTRF was more effective than mTRF at improving insulin sensitivity. Furthermore, eTRF, but not mTRF, improved fasting glucose, reduced total body mass and adiposity, improved inflammation, and increased gut microbial diversity. No serious adverse events…discussion”; Why are the results on BMI from this study not included in the Forest Plot in Figure S2? How do the eTFR compare with the control group?; Why the results relating to this study on total cholesterol, HDL, LDL and triglycerides were not analyzed/inserted in Figures S5 to S8; and the same applies to systolic and diastolic pressure values.   -Zheng et al (2019): these authors state “In the every-other-day fasting group, fasting lasted from 09:00 P.M. on day 1 to 06:00 P.M. on the following day (day 2).for patients with spinal cord injury.”; Once again, the heterogeneity of the patients/participants included is highlighted and then whether or not to value for the clinical conclusions (morbidity, age, weight, BMI...)   - Cai et al (2019) (these authors state): “non-alcoholic fatty liver disease (NAFLD); n=271;” the same question again…because the results of the ADF population (Alternate-day fasting; n=95) and only the TRF (n=95) (, time-restricted feeding) were not considered; control (n=79); Why were the data relating to the ADF population not included in the various Forest plots in this study? Comparing with the control (Ad libium diet control group.)Why were the results of systolic and diastolic blood pressure not included in this meta-analysis? (“systolic or diastolic blood pressure at both time points (data not shown).” ..but it would not be acceptable.  Is it possible to retrieve this data and insert it into the current Forest plot?   - Lin et al (2022) (these authors state): “to investigate the effects of time-restricted feeding (TRF) and a traditional weight-loss method on body composition and cardio-metabolic risk factors in middle-aged women”; “Inclusion criteria were female gender, age 40 to 65 y, body mass index (BMI) ≥ 24 kg/m2, waist-circumference > 80 cm, possession of a smartphone, consent (signing the human-subject research agreement form), stable body weight throughout the past 6 mo (BMI ± 2%), non-smoking, consumption of < 140 g/wk of alcohol”; “randomly assigned into a TRF group (limit 8 h of eating time and fasting for 16 h) or a non-TRF group (traditional weight-loss method, unrestricted eating time) for 8 wk”…only female???? patients… (discussion – methodology heterogeneity).

- Liu NEJM (2022) (these authors state) “Participants were eligible if they were 18 to 75 years of age and had a body-mass index (BMI, the weight in kilograms divided by the square of the height in meters) that was between 28 and 45…)… the men were instructed to follow a diet of 1500 to 1800 kcal per day and the women to follow a diet of 1200 to 1500 kcal per day. B” 2) Participants in the time-restricted–eating group were instructed to consume the prescribed calories within an 8-hour period (from 8:00 a.m. to 4:00 p.m.) each day”. Age (31.9±9.1) (discussion clinical heterogeneity)? Methodology (reinforcing aspects related to heterogeneity, which were well identified and evaluated in this article under analysis).   - Lines 148-155 - “…Heterogeneity was described as moderate (I2>30%), substantial (I2>50%) and considerable (I2>75%). For studies identified to have high heterogeneity, double-checking on input data was performed [14]. Sensitivity analysis was completed by excluding each included study one by one and re-calculating the pooled estimate of the remaining studies, so that studies that largely affected the summary effect or heterogeneity were identified. For studies with multiple IF interventions, we have recalculated effect estimates using the least commonly adopted intervention arm.”. Correct methodology.

Heterogeneity – Results The authors correctly explain this “problem” throughout the text (and the Figures cited). -       Figure S1 (Forest plot for differences in body weight (in Kg): RCTs Ad Libitum diet control group: I2 = 76% (considerable heterogeneity) and for Caloric restricted diet control group I2 = 0% (homogeneous); but -1.40 Kg ( p< 0.00001) in weight after Intermittent Fasting does it have any clinical significance (to discussion)?... Test for overall effect: p< 0.0001); but can we draw practical conclusions for clinical recommendations with such marked heterogeneity (I2 = 89.1%)?   - Figure S2 (Forest plot for differences in body mass index (in Kg/m2): RCTs Ad Libitum diet control group: I2 = 91% (considerable heterogeneity); Test for subgroup differences: p< 0.002; I2 = 89.1% (but can we , again, draw practical conclusions for clinical recommendations with such marked heterogeneity, even if the BMI decreased by 0.55 Kg/m2 with the Intermittent Fasting intervention??) (Total: I2 = 93%; Test for subgroup differences I2 = 79.0 %) compared to the two forms of calorie restriction…. -The same can be said for the content of Figure S3 (Waist circumference): can we draw practical conclusions for clinical recommendations with such marked heterogeneity, even if the Waist Circumference has decreased by 2.12 cm with the Intermittent Fasting intervention??) (Total : I2 = 83%; Test for subgroup differences I2 = 79.3%) comparing with the two forms of caloric restriction…

The essential question that remains, after analyzing all the results presented, is whether the considerable heterogeneity presented allows us to recommend IF to achieve the therapeutic objectives indicated.   Lines 251- 276: Safety (Table 3) and Sensitivity analysis -no questions   Discussion - Lines 291-297: very good summary of main findings and also Potential mechanisms of different IF strategies (although discussion shortened due to lack of space? See below possible mechanisms underlying the benefits achieved - Lines 313-315: the authors state “From this meta-analysis, IF was useful in weight loss. However, it was not beneficial to some cardiometabolic parameters, such as SBP and DBP. A previous meta-analysis with 314 694 participants also showed similar findings, in which TRE could significantly reduce …” “Therefore, further studies should be conducted to verify the effects of IF on blood pressure among Chinese population.” And with different protocols in terms of IF and its duration; I must say… for example; Wilhelmi de Toledo et al (2019) “showed” (Wilhelmi de Toledo F, Grundler F, Bergouignan A, Drinda S, Michalsen A (2019) Safety, health improvement and well-being during a 4 to 21-day fasting period in an observational study including 1422 subjects. PLoS ONE 14(1): e0209353. https://doi.org/10.1371/journal. pone.0209353), report on an observational study including 1422 individuals (Buching fasting therapy – 4 to 21 days fasting…“ Baseline values of systolic blood pressure (SBP) and diastolic blood pressure (DBP) were higher in the groups fasting longer (Fig 3A and 3B). The mean values for the whole cohort decreased significantly from 131.6±0.7 to 120.7± 0.4 for SBP (fasting intervention: p<0.01, and from 83.7±0.4 to 77.9±0.3 for DBP (fasting intervention: p<0.001). The reduction of SBP and DBP was greater in the groups who fasted longer (fasting duration group- by-fasting intervention:...stabilizing for the whole cohort around 120/78 mm Hg…”.

- Can the results of the present meta-analysis on blood pressure be explained by the short duration of the RCTs? To explain the reduction in blood pressure, Wilhemi de Toledo invokes some mechanisms…but in longer words (“Blood pressure reduction might be triggered by factors such as the increase of parasympathetic activity due to the release of brain-derived neurotrophic factor (BDNF) [ 2, 41, 42], increased renal Na excretion [43] and enhanced receptor sensitivity of natriuretic peptides and insulin [44]. more recently in smaller studies on Buchinger fasting [33, 34] and water fasting [48]–also described this blood pressure-reducing effect.”): this can enrich the discussion…. - The same can be questioned regarding the non-reduction of total cholesterol, HDL, triglycerides and glucose concentrations. As per the example cited in the aforementioned publication by Wilhemi de Toledo et al (2019) ---with different protocols in terms of IF and its duration? (“We further found decreases in blood lipid levels following the fasting periods: TG levels as well as TC and LDL-C decreased significantly in all groups. Glucose levels and HbA1c also decreased significantly which points out to a positive effect of fasting on glucoregulation)” again… to improve the discussion there are good explanations in the same article (“…BDNF, associated with neurogenesis and neuron protection… reduction of insulin and leptin … Endogenous opioid (β-endorphin) …  increase in urinary ketone bodies excretion….”).   -Lines 333-337: the authors state “current meta-analysis had pooled the effects from multiple studies conducted in China and had shown statistically significant overall reductions in body weight, BMI and TG when compared to calorie-restricted diet”. and also when IF is compared to the Ad libitum diet…” While more large-scale studies with longer study duration should be conducted, current findings demonstrated the potential clinical significance of IF on cardiometabolic health that warranted in-depth investigation”; try to find a pathophysiological explanation.   Lines 366 – 376: The authors state “4.6. Limitations of the study. … First, most included RCTs included had small number of recruited subjects and short follow-up period….. IF and calorie-restricted diet has not been comprehensively assessed… considerable heterogeneity was observed in multiple meta-analytic results, which may limit the interpretation accuracy… The between-study differences, such as the adherence to fasting regimens, metabolic status, diet and exercise level of the subjects, which were rarely reported in the included studies, might also contribute to high heterogeneity [40]”; in fact I agree that they can explain the considerable heterogeneity (which neither the sensitivity analysis nor the subgroup analysis mitigated in an important way), appear to be well carried out in the current study… but also the morbidities… the duration of the studies…..   Lines 377 – 384: excellent topic – Conclusions   References: apparently few (n=40) but very current and well integrated with the theme under analysis. From a formal point of view, it respects the norms required by the Nutrients- MDPI/Scientific Publication.     The results are what they are. The initial question was well posed: “previous literature suggested a lack of systematic review of the efficacy of IF on improving cardiometabolic health of Chinese population, and whether the potential benefit of IF is independent of calorie restriction during intervention. Through gaining additional understanding on the efficacy IF in Chinese population, it may help healthcare professionals to formulate evidence-based weight loss plans for individuals with overweight and obesity.” However, the marked heterogeneity already mentioned may raise some questions: 1) The authors found RCTs with very different characteristics among participants…age…comorbidities…type of Intermittent fasting…duration of IF (perhaps 3 weeks would be too short to obtain more significant differences and with greater clinical importance…); Therefore, Table S2 should include a column with the profile of the participants (to explain one of the reasons for the considerable high heterogeneity found).   As already mentioned, this article is well written and meets the requirements of a Meta-Analysis study. The different chapters are correctly prepared from a scientific point of view (Introduction, Material and Methods, Results... Discussion, Limitations, Conclusions). The discussion is well built and according to the results obtained. But it can be enriched with possible pathophysiological explanations and compared with results that are not similar published by other authors and even explanations of the possible benefits found in other studies (even in an Asian population), and above all better explain the considerable heterogeneity found for so many important results taking into account the study objectives; this aspect must be improved; It must also be explained why some of the aspects referenced for each study included in this meta-analysis were not considered – see above). The limitations are well outlined and the conclusions agreed on the basis of the results obtained. The references are apparently few but very current and well integrated with the theme under analysis. From a formal point of view, it respects the norms required by the Nutrients/MDPI Scientific Publication.

Comments on the Quality of English Language

Reviewer 2 Report

Comments and Suggestions for Authors

The discussion needs improvements:

The authors need to report other type of fasting mentioning also long term fasting (Ann Med. 2020 Aug;52(5):147-161)

Why does the intermittent fasting reduce insulin and not glucose? Please discuss the concept of insulin resistance (Cienfuegos et al., 2020, Cell Metabolism 32, 366–378 and many more

Why does the intermittent fasting reduce LDL-C and not HDL-C? (The Journal of Clinical Endocrinology & Metabolism, Volume 106, Issue 3, March 2021, Pages e1468–e1470 and many more

Why does the intermittent fasting reduce BMI and not waist circumference?

Why did the author refer to American guidelines? Please delete.

Comments on the Quality of English Language

Must be improved

Round 2

Reviewer 1 Report

Comments and Suggestions for Authors The authors of the article proposed for publication responded correctly and adequately to the questions listed regarding the first version and included new meaningful excerpts and several new appropriate citations (an additional 22 citations), and added an assertive review of the results and enriching expansion of the discussion. Therefore, I am of the opinion that this new version should be accepted as formulated. By way of example and responding in particular to some questions by the authors: The authors highlight the excerpts from the first reviews (in red what was written as a reviewer) and give their response: As already mentioned, Table S2 will be enriched if it has another column with the characteristics of the participants included”. - Response: Thank you for your suggestion. We have inserted the column of participants’ comorbidities in Table S2. In the same table, we have also described the age, % of illnesses and baseline BMI of participants”. Correct insertion of a new column covering the comorbidities of patients included in the different clinical trials and which will be published in the main text (I hope). - Lines 50-51: new version “There are various forms of fasting developed, including intermittent fasting (IF), calorie restriction, long-term fasting, very low-calorie diet and fasting mimicking diet [6].” Correct statements and also the new citation.   Line 168: “and one has evaluated ADF [23]”. According to Chair et al (2022) [17]: “The participants were randomized into the ADF group (n = 34), 16/8 TRF group (n = 33), and control group (n = 34). Therefore, there are 2 studies that evaluated ADF and not just one, that of Zheng et al (2021) [23] Response: Thank you for your careful review. We have updated the number in the first paragraph of Section 3.2. Study characteristics. – Column correctly inserted. Response: Thank you for your question. The 4 included studies have used different methods to assess body fat composition, as listed below: (i) Cai et al. 2019: Body composition (fat mass and fat-free mass) was assessed by dual x-ray absorptiometry (DXA, Discovery-W version12.6, Hologic, Bedford, MA, USA). (ii) Liu et al. 2022: The body fat mass and lean mass were quantified with the use of a whole-body dual radiography system (Lunar iDXA, GE Healthcare). (iii) Xie et al. 2022: Body mass and percentage body fat were measured using an HBF-371 Bioelectrical impedance analyzer (Omron Healthcare Co., Kyoto, Japan). (iv) Guo et al. 2021: Weight, percentage body fat, and visceral fat index were measured with an Innerscan Segmental Body Composition Monitor (No. 570509, Tanita Corporation). We agree that the differences in assessment methods could be a potential factor leading to the heterogeneity. However, with reference to Figure S4, the consistency of results was not affected despite the differences in body fat assessment methods, as reflected by 0% I-square heterogeneity. Thus, we have stated this observation in the revised manuscript (Section 3.4. Effect on body fat composition), “The consistency of results was not affected despite the differences in body fat assessment methods”. – I agree with the authors’ comments.   Line 204-232: _ New correct and amended version: “Eight studies involving 712 participants…Effect on blood lipid profile…Effect on blood glucose…” with several new citations that are appropriate from [35 to [44]….” Text is added in a correct and appropriate way from lines 307 to 371 “Similar to results from previous studies, IF did not lead to significant change in blood … vital importance to verify the health effects of IF among Chinese population.”.   Line 372: “4.2. Potential mechanisms of different IF strategies” longer text with relevant content and in accordance with the authors’ new proposal “We have revised Section 4.2. Potential mechanisms of different IF strategies”.   We have added the explanation of choosing eTRE in Section 3.10. 5. Lines 173-230: “Response: Thank you for your comment. We have added the clinical significance of our findings and described how heterogeneous of the participants were and its potential effects on the results in paragraph 5 of Section 4.1. Summary of main findings”. and in fact they responded appropriately.   6. From reading the articles included in the Meta-Analysis (indicated below), some questions/suggestions emerge. Therefore, the studies highlight the following aspects (in red).) “Response: Thank you for your comment. We have updated Table S2 accordingly. Among the 9 included studies, 6 of them studied the efficacy of TRE. Therefore, TRE was selected from multi-armed studies for meta-analysis to increase the comparability across studies. Instead of excluding the data of ADF group, in the updated sensitivity analysis, we have used it to replace the data of TRE group and re-run the meta analysis. We have added Table S5 to show the results. There was no substantial difference observed in terms of statistical significance nor direction of intervention effects in most parameters.” Adequate correction to what was requested, as well as the answers to the questions formulated for the different studies included in the meta-analysis

Comments on the Quality of English Language

Minor editing of English language required

Author Response

We thank the reviewer for the careful and insightful review of our manuscript.